# Pressure-induced ferroelectric-like transition creates a polar metal in defect antiperovskites $Hg_3Te_2X_2$ (X = Cl, Br)

Weizhao Cai [1], Jiangang He[2✉], Hao Li[3], Rong Zhang[1], Dongzhou Zhang [4], Duck Young Chung [3], Tushar Bhowmick[1], Christopher Wolverton [2], Mercouri G. Kanatzidis[3,5✉] & Shanti Deemyad [1✉]

Ferroelectricity is typically suppressed under hydrostatic compression because the short-range repulsions, which favor the nonpolar phase, increase more rapidly than the long-range interactions, which prefer the ferroelectric phase. Here, based on single-crystal X-ray diffraction and density-functional theory, we provide evidence of a ferroelectric-like transition from phase $I2_13$ to $R3$ induced by pressure in two isostructural defect antiperovskites $Hg_3Te_2Cl_2$ (15.5 GPa) and $Hg_3Te_2Br_2$ (17.5 GPa). First-principles calculations show that this transition is attributed to pressure-induced softening of the infrared phonon mode $\Gamma_4$, similar to the archetypal ferroelectric material $BaTiO_3$ at ambient pressure. Additionally, we observe a gradual band-gap closing from ~2.5 eV to metallic-like state of $Hg_3Te_2Br_2$ with an unexpectedly stable $R3$ phase even after semiconductor-to-metal transition. This study demonstrates the possibility of emergence of polar metal under pressure in this class of materials and establishes the possibility of pressure-induced ferroelectric-like transition in perovskite-related systems.

[1] Department of Physics and Astronomy, University of Utah, Salt Lake City, UT, USA. [2] Department of Materials Science and Engineering, Northwestern University, Evanston, IL, USA. [3] Materials Science Division, Argonne National Laboratory, Lemont, IL, USA. [4] PX2, Hawaii Institute of Geophysics and Planetology, University of Hawaii at Manoa, Honolulu, HI, USA. [5] Department of Chemistry, Northwestern University, Evanston, IL, USA. ✉email: jiangang2020@gmail.com; m-kanatzidis@northwestern.edu; Deemyad@physics.utah.edu

ABO$_3$ perovskites represent a significant class of multifunctional materials, possessing a broad range of applications in electronics, such as transistors, tunable capacitors, and nonvolatile memories due to their intriguing physical properties[1]. Their frameworks which consist of A cations encapsulated in corner-sharing BO$_6$ octahedra and their stabilities as function of pressure and temperature have extensively been investigated[2–4]. As first noticed by Samara et al. 45 years ago, and supported by majority of experiments, hydrostatic pressure usually suppresses ferroelectric distortion in these perovskites[5–7]. This effect which is mainly attributed to a faster increase in short-range interactions than long-range interactions under pressure has been observed in many archetypal ferroelectric compounds, including BiFeO$_3$ and BaTiO$_3$ (refs. [8–10]). Therefore, negative pressure or strain is frequently used to enhance ferroelectricity[11,12].

The prototypical perovskite PbTiO$_3$ was theoretically predicted to be the first case that violates Samara's theory. First-principles calculations show that the ferroelectricity of PbTiO$_3$ is first suppressed at a critical pressure, but then is unexpectedly enhanced by the further compression[13]. However, experimental investigations found that PbTiO$_3$ undergoes complex structural phase transitions with elevating pressure: the ferroelectric phase $P4mm$ is first suppressed by pressure (at ~13 GPa) and it then reenters into a polar phase $I4cm$ from a nonpolar phase $I4/mcm$ at 45 GPa (ref. [14]). Attempts to search for pressure-induced ferroelectricity in other ABO$_3$ materials have also remained inconclusive. For example, Guennou et al. reported a nonpolar-to-polar transition in BiMnO$_3$ at 50 GPa based on the rietveld refinement of the powder X-ray diffraction (XRD) data[15]. Considering the structural complexity and the difficulty of determining the ground structure of BiMnO$_3$ (refs. [16–19]) and the limited data available at such high pressure, the assignment of the noncentrosymmetric (also monoclinic) structure induced by pressure remains obscure[20]. Therefore, the experimental evidence of a direct pressure-induced ferroelectric-like phase transition in perovskites and related materials remained elusive up to this point.

Ferroelectricity conventionally emerges in semiconductors and insulators, and since the conduction electrons screen out the static internal field generated by dipole moment, it is not typically expected to be present in metals. The possibility of a ferroelectric-like transition in metals was first proposed by Anderson and Blount in the 1960s (ref. [21]). However, until the recent report of the "ferroelectric polar metal" in LiOsO$_3$, which displays nonpolar ($R\bar{3}c$) to polar ($R3c$) structural transition at 140 K and ambient pressure, no ferroelectric phase transitions in any metals have been reported[22]. Further theoretical study shows that the nonpolar-to-polar transition temperature in LiOsO$_3$ is slightly increased by pressure in the low-pressure range and the polar phase $R3c$ is ultimately suppressed at 21 GPa (ref. [23]). Recently, ferroelectricity was reported in bulk crystalline two-dimensional WTe$_2$ at ambient conditions. The emergence of the ferroelectricity in this Weyl semimetal could be correlated to its layered structure together with strong electronic anisotropy[24]. However, examples on the coexistence of metallic and polar states under high pressure in perovskites and related materials are rarely identified[25].

Here, by combining experimental and computational studies, we present evidence of a pressure-induced ferroelectric-like transition in defect antiperovskites Hg$_3$Te$_2$X$_2$ (X = Cl and Br) at moderate pressures (10.5–17.5 GPa). Moreover, we find further compression leads to a transition to polar and metallic states. Our findings not only present rare but unambiguous examples of a direct nonpolar-to-polar transition promoted by hydrostatic pressure, but also open a door to explore new ferroelectric/multiferroic materials by employing external pressure.

Antiperovskite, is a structure similar to the perovskite structure with inverted cations and anions. The known examples of inorganic materials with antiperovskite structure include alkali metal-rich compounds Li$_3$OX (X = Cl, Br)[26,27], carbides such as Mn$_3$GaC[28] and defect antiperovskites K$_2$Se$_2$O$_3$ (ref. [29]), Fe$_2$SeO[30], and Cs$_3$Sb$_2$I$_9$ (ref.[31]). In comparison with the conventional ferroelectric perovskites, the pressure behavior of antiperovskites remains poorly explored[32–34]. The ambient-pressure phases of antiperovskites are typically robust and stable in a wide pressure range (e.g., Na$_3$OBr (ref. [34])). In addition, a few organic–inorganic antiperovskites have recently been reported. For example, [(CH$_3$)$_3$NH]$_3$(MnBr$_3$)(MnBr$_4$) shows a ferroelectric-to-paraelectric transition at a high temperature of 458 K at ambient pressure[35]. In contrast to the inorganic antiperovskites, this type of framework can easily collapse under moderate pressures.

Mercury chalcohalides, Hg$_3$Te$_2$X$_2$ (X = Cl, Br) are wide band gap semiconductors ($E_g$ ~2.5 eV) at ambient pressure, and they belong to a chiral noncentrosymmetric, nonpolar space group $I2_1 3$ (point group 23). Their three-dimensional (3-D) structure can be regarded as antiperovskite type with ordered mercury vacancies[36]. They have potential applications in hard radiation detectors, piezoelectric, and nonlinear devices, but so far not in ferroelectrics[37,38].

In this work, using synchrotron XRD on both single crystal and powder samples of defect antiperovskites Hg$_3$Te$_2$X$_2$ (X = Cl, Br), we observe a pressure-induced direct ferroelectric-like transition to a polar space group $R3$ (point group $C_3$). Moreover, we found the critical pressure of the transition is highly sensitive to hydrostatic environments. The polar phase of Hg$_3$Te$_2$Br$_2$ initially maintains its semiconducting property after the phase transition, and with further compression, we observe a complete band gap closure and a semiconductor-to-metal transition at 42.3 GPa. Hence, a polar metal is observed at high density at room temperature. Density functional theory (DFT) calculations reveal the phase transition is a direct consequence of the softening (negative frequency at high pressure) of the polar phonon mode $\Gamma_4$ under pressure, consistent with group theory analysis that shows $I2_1 3$ to $R3$ is a proper ferroelectric phase transition.

## Results

**Structural evolution and phase transitions in Hg$_3$Te$_2$X$_2$ under pressure.** The crystal structures of Hg$_3$Te$_2$X$_2$ under pressure were determined by synchrotron single-crystal XRD at room temperature. To ensure a good hydrostatic environment and investigate the effects of quasi-hydrostatic stress on the phase stabilities of Hg$_3$Te$_2$X$_2$, both helium (He) and methanol–ethanol–water (MEW) were employed as pressure-transmitting media (PTM) in different runs. In addition, to examine the crystal structures at identical conditions to electrical transport measurements, both compounds were also studied by powder XRD without PTM. As shown in Fig. 1, the ambient-pressure noncentrosymmetric phase (space group $I2_1 3$, here denoted as phase I) persists up to ~15.5/17.5 GPa for Cl/Br, when He is used as the PTM. The pressure induces a discontinuous phase transition to a new phase II, a polar phase, in the rhombohedral system, with the space group $R3$ (a subgroup of $I2_1 3$ phase, obeys the Curie symmetry principle), which belongs to one of the ten polar point groups, $C_3$ (ref. [39]). The unconventional symmetry change from chiral ($I2_1 3$) to chiral ($R3$) can be described by the Aizu notation of 23F3, among 22 species of chiral-to-chiral ferroelectric phase transitions[40,41]. Normally, for conventional ferroelectric transitions at ambient pressure, the high-temperature paraelectric phase (centrosymmetric) transforms to the low-temperature ferroelectric phase (noncentrosymmetric) upon cooling, which is associated with symmetry breaking and establishment of electron

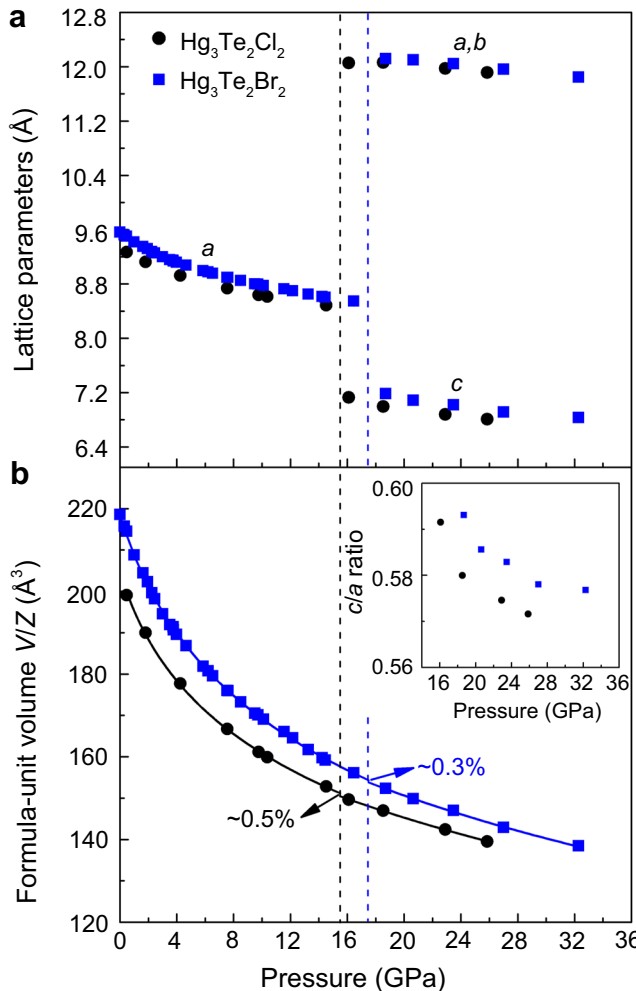

**Fig. 1 Lattice parameters of Hg₃Te₂X₂ compressed in helium at room temperature. a** Lattice parameters $a$, $b$, and $c$ as a function of pressure obtained from single-crystal X-ray diffraction. **b** The third- and second-order Birch–Murnaghan equations of states (EOS) fit to the formula-unit volume ($V/Z$) data. These EOS have been used for calculating the $\Delta V$ collapse at the phase transition between phases I and II. The calculated bulk moduli are given in Supplementary Table 1. The inset in **b** indicates the $c/a$ ratio of phase II in both compounds. Vertical black and blue dashed lines indicate phase transitions at 15.5 and 17.5 GPa for X = Cl and Br, respectively. The error bars are smaller than the symbols used in both **a** and **b**.

polarization[42]. Although, ferroelectric transitions involving symmetry change from noncentrosymmetric-to-noncentrosymmetric do exist, but, only a few examples of hybrid perovskites such as [NH₄][Zn(HCOO)₃] are reported to possess such type of transition[43]. It is even more uncommon to find an example of displacive ferroelectric phase transition, where the transition occurs from a noncentrosymmetric–nonpolar structure to a noncentrosymmetric–polar structure. The first-order transformation observed here is marked by a subtle collapse in the formula-unit volume ($V/Z$) by ~0.5%/0.3% and distinct changes of the XRD patterns (Fig. 1 and Supplementary Fig. 1). Notably, the phase transitions occur at lower pressures of ~10.5/11.0 GPa for Cl/Br when MEW is used as the PTM, and an isostructural transition to a new phase II′ takes place at ~14.4/15.0 GPa (the lattice nodes and space group remain the same, but with discontinuous changes of lattice parameters and $c/a$ axial ratio, see Supplementary Figs. 2 and 3). The absence of the volume collapse

suggests the possibility of a second-order phase transition. The first phase transition of Hg₃Te₂Cl₂ takes place at the same critical pressure both in the absence of PTM and when MEW is used as PTM (~10.5 GPa). However, in the absence of PTM, the II–II′ transformation at higher pressures is not observed (Supplementary Figs. 2 and 3) and phase II survives to our experimental limit of 38.1 and 49.0 GPa for Hg₃Te₂Cl₂ and Hg₃Te₂Br₂, respectively (Supplementary Fig. 2). Upon decompression, the high-pressure phases could be completely recovered to the ambient-pressure phase I regardless of the PTM (Supplementary Tables 2 and 3). It can be easily seen that the onset of phase transitions of Hg₃Te₂X₂ strongly depends on stress and major differences exist in high-pressure phase boundaries due to different hydrostatic conditions. For example, at room temperature MEW only provides good hydrostatic condition up to 10.5 GPa, whereas He is a perfect hydrostatic medium, and even maintains nearly perfect hydrostaticity >20 GPa (ref.[44]). The strong dependence of phase transitions on hydrostaticity has been observed previously in other systems, including the archetype perovskite BiFeO₃ (ref. [45]).

The compressibility of the lattice parameter $a$ of Hg₃Te₂X₂ in He and MEW are very similar within the phase I region (e.g., 6.2 (2) vs. 6.3 (1) TPa⁻¹ in Hg₃Te₂Br₂). Notably, phase II in MEW exhibits an unusual negative area compressibility (NAC) behavior (expands within the $ab$ plane under pressure), which continues monotonically to ~14.4/15.0 GPa for Cl/Br (Supplementary Fig. 3), whereas phase II in He exhibits normal positive compressibility, at least up to the experimental limit of 25.9/32.3 GPa for Cl/Br. The abnormal NAC feature in MEW most likely originates from the pseudo-hydrostatic environment, in which deviatoric stress leads to nonuniform compression of the crystal structure[46].

At ambient conditions, the Hg₃Te₂X₂ compounds adopt the corderoite mineral ($\alpha$-Hg₃S₂Cl₂) structure type, with one Hg, one Te and one halogen (X) atoms in the asymmetric unit[47]. Their 3-D structures are constructed from an infinite spatial framework, which is composed of interconnected TeHg₃ pyramids, and the halogen atoms reside in the cavities (Fig. 2). However, based on the topological feature of the structures, if we consider the TeHg₃ pyramids as part of the TeHg₃□₃ octahedra (where □ represents a vacancy), the structure can be treated as a defect antiperovskite with a formula of Hg₁.₅□₁.₅TeX, in which 50% of the Hg atoms are missing (Fig. 2a)[36]. As pressure increases, TeHg₃□₃ octahedra become more distorted, and the layers of halogen atoms within free spaces change from roughly flat to significantly corrugated (Supplementary Fig. 4). The Hg1 atom adopts a distorted seesaw geometry, which is coordinated to two Te1 and two halogen atoms (Supplementary Fig. 6a). The evolution of Hg1–Te1 and Hg1–X1 bond distances, and Te1–Hg1–Te1$^i$ angles as a function of pressure in MEW and He are well consistent (Supplementary Fig. 6). Notably, the Hg1–Te1 distances in the Hg₃Te₂X₂ are very rigid in response to pressure, i.e., the Hg1–Te1 bond lengths, which are 2.6599(6)/2.6667(7) Å for Cl/Br at 0.1 MPa are changed only to 2.6425(26)/2.6771(17) Å when pressure increased to 14.5/16.5 GPa within phase I region in He (Supplementary Fig. 6b, c). The Te1–Hg1–Te1$^i$ tilt angle within two defect [TeHg₃□₃] octahedra manifests considerable changes under compression, e.g., they increase initially with pressure up to ~5.2/4.3 GPa, and then decrease under further compression (Supplementary Fig. 6d, e). In Hg₃Te₂Br₂, the Hg1–Br1 distances are equal to 3.0713(7) Å at 0.1 MPa, which are considerably larger than the corresponding covalent radii (2.62 Å). This enlarged interatomic distance suggests the presence of weak ionic bonding in the structure. With increasing pressure to 16.5 GPa, the Hg1–Br1 lengths decrease to 2.777(3) Å (Supplementary Fig. 6g).

In the high-pressure phase II, there are two independent Hg atoms (Wyckoff sites, 9$b$ and 9$b$), two Te atoms (Wyckoff sites,

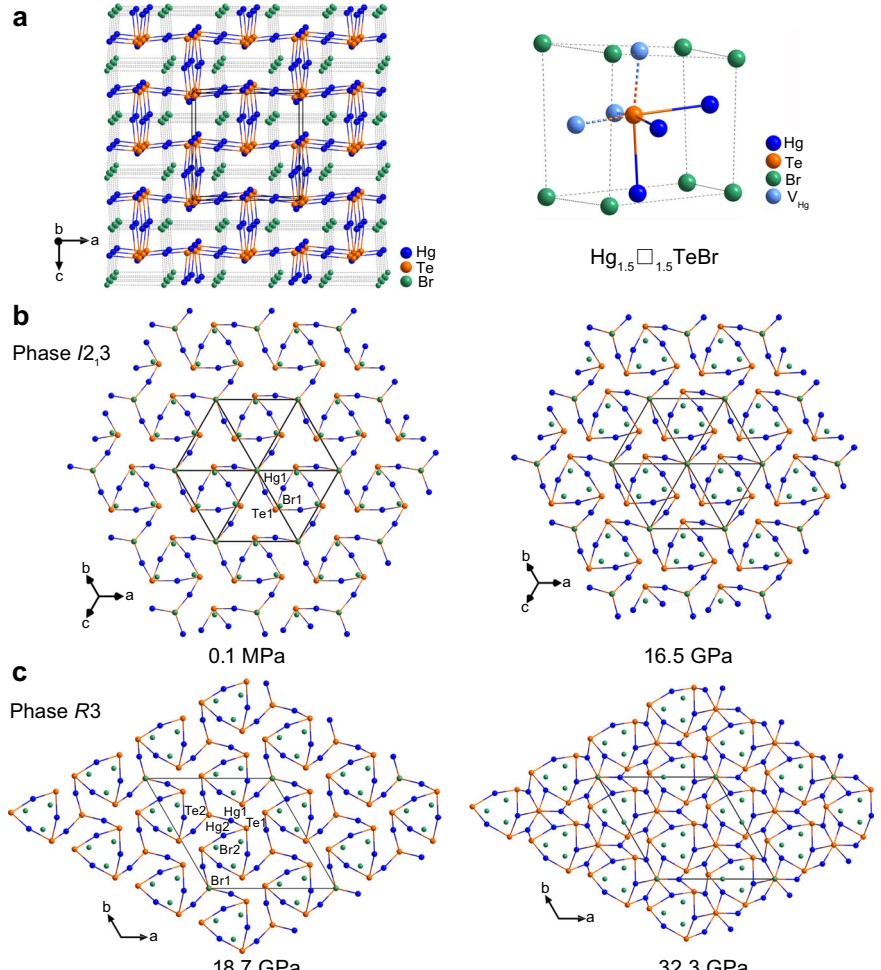

**Fig. 2 Crystal structures of Hg₃Te₂Br₂ at different pressures. a** Structure of phase I of $Hg_3Te_2Br_2$ viewed approximately along the [010] direction at 0.1 MPa. The defect antiperovskite structure of $Hg_{1.5}\square_{1.5}TeBr$, with 50% of Hg atoms missed (perovskite-lime pseudo-cage shown). **b** Phase I at 0.1 MPa (left) and 16.5 GPa (right) viewed along the [111] direction. **c** Phase II at 18.7 GPa (left) and 32.3 GPa (right) viewed along the c-axis. The coordination numbers of Te1 and Te2 atoms increase under pressure are shown in 32.3 GPa (Supplementary Fig. 5). All the Hg–Br bonds are omitted for clarity.

9b and 3a), and two halogen X atoms (Wyckoff sites, 9b and 3a) in the asymmetric unit. As shown in Fig. 2c, the 3-D framework of phase II retains the main architecture of phase I, and the structure is composed of two different types of [TeHg₃] pyramids. These pyramids are interlinked via vertex-sharing and the halogen X atoms are located in free spaces. Therefore, the symmetric changes from phases I to II can be simply understood by the removal of the symmetry elements of twofold rotation and screw axes (Fig. 2b, c). Like in phase I, the Te1 and Te2 atoms feature triangle pyramid-like geometry, and their coordination numbers increase from 3 to 4 and to 8 when pressure is increased to 32.3 GPa (Supplementary Fig. 5). Except for the Hg1–Te2 bond, other Hg–Te distances are reduced only by a small amount under compression in both MEW and He (Supplementary Fig. 6). It is notable that one type of the Te–Hg–Te angles in phase II shows a continuous reduction during and after the transition from ambient-pressure phase I, while another one shows a considerable and abrupt decrease during the phase transitions (Supplementary Fig. 6d, e). In contrast to phase I, the coordination environment of Hg atoms in phase II are quite different. Both Hg1 and Hg2 atoms adopt a distorted octahedral geometry and connect together via face-sharing (Supplementary Fig. 6a) and all the Hg–X bond lengths are slightly reduced when the pressure is increased to 32.3 GPa.

**Pressure-induced semiconductor-to-metal transition of Hg₃Te₂Br₂.** The $Hg_3Te_2Cl_2$ and $Hg_3Te_2Br_2$ are wide band gap semiconductors with energy gaps of 2.6 and 2.5 eV at ambient pressure, respectively[48]. When the single crystals were pressurized in He, we observed their colors gradually changed from yellow to dark orange and then to opaque black >10 GPa (Supplementary Fig. 7). During the decompression, we find that these changes are completely reversible, and the dark color reverted back to the original yellow color. These observations are well consistent with XRD measurements as discussed above. The prominent piezo-chromism of $Hg_3Te_2X_2$ reveals considerable electronic changes under compression and prompted us to investigate their electrical transport properties. At ambient conditions both $Hg_3Te_2X_2$ samples have high resistivity (>10⁹ Ω·cm), and up to ~8.0 GPa their electrical resistivities fall beyond the range we could measure[48]. With increasing pressure, both compounds transform to the high-pressure polar phase R3, in which their electrical resistances are still very large (~10⁷–10⁸ Ω), indicating semiconducting character. Further compression leads to the electrical resistances falling drastically by more than nine orders in magnitude for pressures >30.0 GPa (Fig. 3a). Similar to the crystal structure and optical properties, the electrical resistances return to their initial value during gradual decompression (Fig. 3 and Supplementary Fig. 8a). In order to confirm the onset of metallization and the presence of metallic state, we examined the

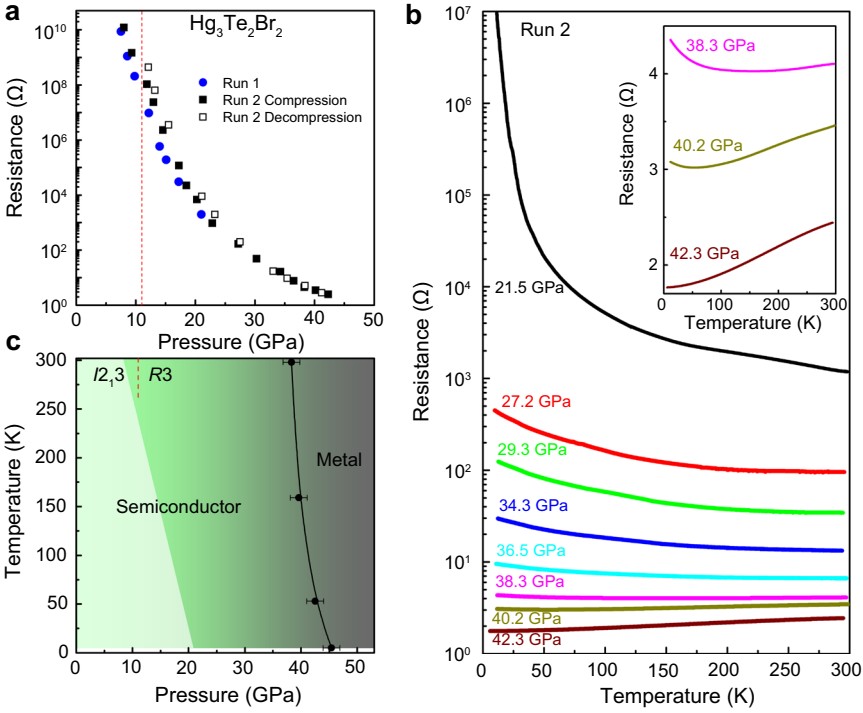

**Fig. 3 Pressure-induced metallization of Hg₃Te₂Br₂.** **a** Pressure-dependent electrical resistances of Hg₃Te₂Br₂ at room temperature. **b** Temperature dependence of the electrical resistance of Hg₃Te₂Br₂ at different pressures. The inset enhances the semiconductor-to-metal transition at 42.3 GPa. **c** P–T phase diagram of Hg₃Te₂Br₂ on the basis of resistance measurements. Solid line: semiconductor–metal phase boundary.

temperature dependence of the electrical resistance of $Hg_3Te_2Br_2$ >21.0 GPa (Fig. 3b) and observed a gradual transition from semiconductor-to-metal under pressure. As shown in Fig. 3b, the $R(T)$ curves show negative $dR/dT$ when pressure is <36.5 GPa in the range of 5–300 K, indicative of semiconducting character. Between 38.3 and 40.2 GPa, the $dR/dT$ displays positive in the high temperature region, whereas it changes to negative in the low-temperature region (e.g., 53 K for 40.2 GPa). With increasing the pressure >42.0 GPa, a positive $dR/dT$ is observed throughout all temperature range (5–300 K), implying the complete metallization of $Hg_3Te_2Br_2$. We used variable range hopping (VRH) model $R(T) = R_0 \exp(T_0/T)^{1/n}$ (where $R_0$ is a characteristic temperature and $n$ is an integer (1–4) depending on the conduction mechanism) to analyze the resistance data <40.2 GPa (ref. [49]). We found the 3-D VRH model ($n = 4$) provides the best fitting to the resistance data <20 K for all pressures, revealing the carrier conduction is dominated by the 3-D VRH mechanism (Supplementary Fig. 8b). In order to analyze the metallic state, we tried to apply the power-formula $R(T) = R_0 + AT^n$ to fit the resistance data at 42.3 GPa <70 K. We observed $Hg_3Te_2Br_2$ shows a slight deviation from ideal Fermi liquid (FL) behavior ($n = 2$) and for this pressure the $R(T)$ data can be best fitted for $n = 1.8$. Pressure-induced non-Fermi liquid (NFL) to FL transition has been observed in strong correlated systems[50]. However, $Hg_3Te_2Br_2$ is not a strongly correlated compound therefore it is unlikely that the slight deviation is due to NFL behavior. At this pressure range ($P > 42$ GPa), the $R3$ phase is present based on previously discussed X-ray analysis, indicative of polar metal behavior (Supplementary Fig. 2c). In $LiOsO_3$ at ambient pressure a broad peak appears in heat capacity, accompanied by the polar-to-centrosymmetric structural phase transition at 140 K, which presents a direct evidence of a second-order phase transition[22]. Such type of phase transition is also typically evidenced by a distinct anomaly in the electrical resistivity. However, in case of $Hg_3Te_2X_2$, since the polar phase emerges at high pressure, we

were unable to perform similar heat capacity measurements as in $LiOsO_3$. Although we could not find any evidence of anomaly in the $R–T$ curves (Fig. 3b), this does not exclude the possibility of a second-order polar-to-centrosymmetric structural phase transition in these two compounds. Further synchrotron X-ray experiments and heat capacity measurements at variable temperature and pressure are needed to experimentally explore this possibility. Figure 3c shows the $P–T$ phase diagram of $Hg_3Te_2Br_2$ based on the XRD and resistance data. In addition, we measured the photoconductivity of polycrystalline $Hg_3Te_2Br_2$ samples under pressure at room temperature illuminated by 532 nm laser light (Supplementary Fig. 9), and observed a steady increase of the photocurrent in the measured pressure range (1.2–12.1 GPa), consistent with the reduction of the semiconducting gap and the creation of a higher concentration of thermally excited carriers.

**Electronic structures of $Hg_3Te_2X_2$ under pressure.** The electronic structures of $Hg_3Te_2Cl_2$ and $Hg_3Te_2Br_2$ were calculated based on the fully relaxed structures. The calculated ambient-pressure lattice constants are 9.3318 and 9.5866 Å, respectively. We find the features of the ambient-pressure band structures and density of states (DOS) between $Hg_3Te_2Cl_2$ and $Hg_3Te_2Br_2$ are very similar: both are indirect band gap semiconductors with valence band maximum at the middle of H–N line and conduction band minimum at Γ point (Fig. 4a, b). The PBEsol (HSE06) calculated ambient-pressure band gaps are 1.84 (2.81) and 1.73 (2.66) eV for $Hg_3Te_2Cl_2$ and $Hg_3Te_2Br_2$, respectively, with the bromide having a smaller gap due to the smaller electronegativity of Br. In general, PBEsol underestimates the band gaps by ~30%, while HSE06 slightly overestimates the band gaps. The valence band near the Fermi level is quite flat along H–N–Γ, which is mainly due to the $p–d$ coupling between Hg $5d$ and Cl $3p$ (or Br $4p$) and Te $5p$ orbitals[51]. The projected crystal orbital Hamilton population (pCOHP)[52] analysis show there is a strong antibonding

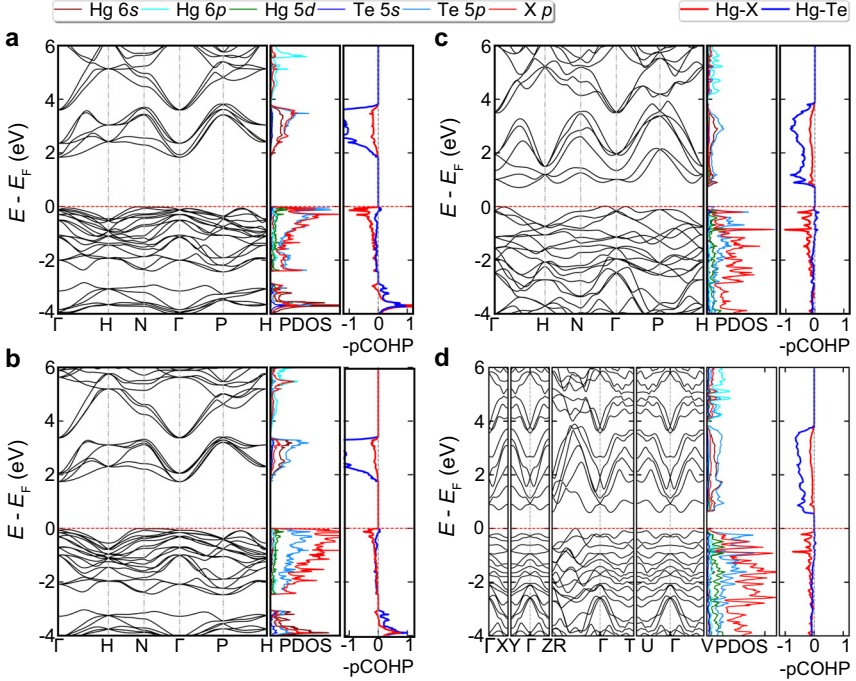

**Fig. 4 Electronic structures of Hg₃Te₂X₂ under pressure.** a Band structures, density of states (DOS), and -pCOHP of $I2_13$ phase of $Hg_3Te_2Cl_2$ at ambient pressure. **b** $I2_13$ phase of $Hg_3Te_2Br_2$ at ambient pressure. **c** $I2_13$ phase (hypothetical) of $Hg_3Te_2Cl_2$ at 20 GPa. **d** $R3$ phase of $Hg_3Te_2Cl_2$ at 20 GPa. Positive and negative -pCOHP indicates bonding and antibonding interactions, respectively.

interaction between Hg 5d and Cl 3p (Br 4p) at the top of the valence band (see Fig. 4 and Supplementary Fig. 10), due to the unusual d–p coupling. The filling of the antibonding levels destabilizes the crystal structure. The more dispersive band around the bottom of conduction bands is mainly contributed by Te 5s and Hg 6s orbitals.

Experimentally, the $I2_13$ phase of $Hg_3Te_2Cl_2$ transforms to $R3$ at 15.5 GPa. In Fig. 4c, d, we show the electronic structures of the ambient-pressure phase $I2_13$ and the high-pressure phase $R3$ at 20 GPa, respectively. Comparing with the electronic structure at ambient pressure (Fig. 4a), pressure remarkably reduces the band band gaps of both $I2_13$ (0.72 eV) and $R3$ (0.60 eV) structures, due to broadening of the valence band and conduction band with shortening Hg–Cl and Hg–Te bond lengths. Meanwhile, the positions of the valence band minimum and conduction band maxima in the Brillouin zone shift away from the H–N line and the Γ point, and the antibonding states are broadening and shift downward.

**Phonon dispersion under pressure.** The primitive unit cell of the $I2_13$ structure has 14 atoms (two formula units, f.u.). Therefore, there are 42 phonon modes at the Γ point: 3 A ($\Gamma_1$) + 3E* ($\Gamma_2\Gamma_3$) + 11 T ($\Gamma_4$). Phonon dispersion of $Hg_3Te_2Cl_2$ as a function of hydrostatic pressure at 0 K is shown in Fig. 5. At ambient pressure, $Hg_3Te_2Cl_2$ shows small soft-phonon modes along the P–H and H–N–Γ directions. When pressure increases to 5 GPa, harder phonon modes can be observed in the whole first Brillouin zone due to the enhancement of the bonding strength, which is a common effect due to reduction of bond lengths. However, with pressure further increased to 20 GPa, $Hg_3Te_2Cl_2$ shows largely unstable (negative frequency) phonon modes at the Γ and H points, with symmetries of $\Gamma_4$ and $H_2H_3$, respectively. The zone center $\Gamma_4$ is a ferroelectric mode, which is connected with three subgroup structures, $C2$, $R3$, and $P1$, along the [100], [111], and [abc] directions, respectively. The zone boundary mode $H_2H_3$ only connects with one subgroup structure $P2_12_12_1$. The atomic

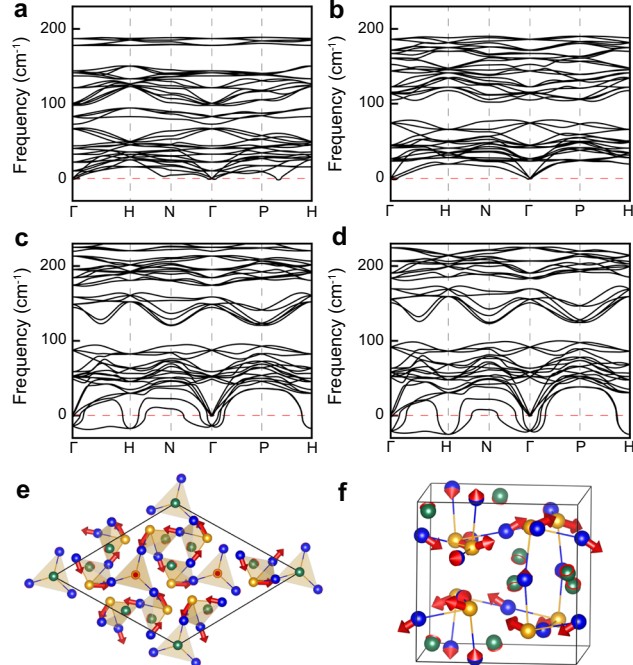

**Fig. 5 Phonon dispersion of $I2_13$ phase of $Hg_3Te_2Cl_2$ under pressure.** **a–d** Phonon dispersion at ambient pressure, 5, 20, and 25 GPa. **e**, **f** Atom displacement of unstable $\Gamma_4$ (polar) and $H_2H_3$ (nonpolar) phonon modes at 20 GPa. Negative frequency indicates an unstable phonon mode. The direction and length of the red arrow represent the displacement direction and amplitude.

displacements of $\Gamma_4$ along the [111] direction and $H_2H_3$ are shown in Fig. 5e, f, respectively. The main distortion of $\Gamma_4$ is from three of four $TeHg_3$ tetrahedra in the unit cell and other $TeHg_3$ tetrahedron nearly does not distort. In the heavily distorted

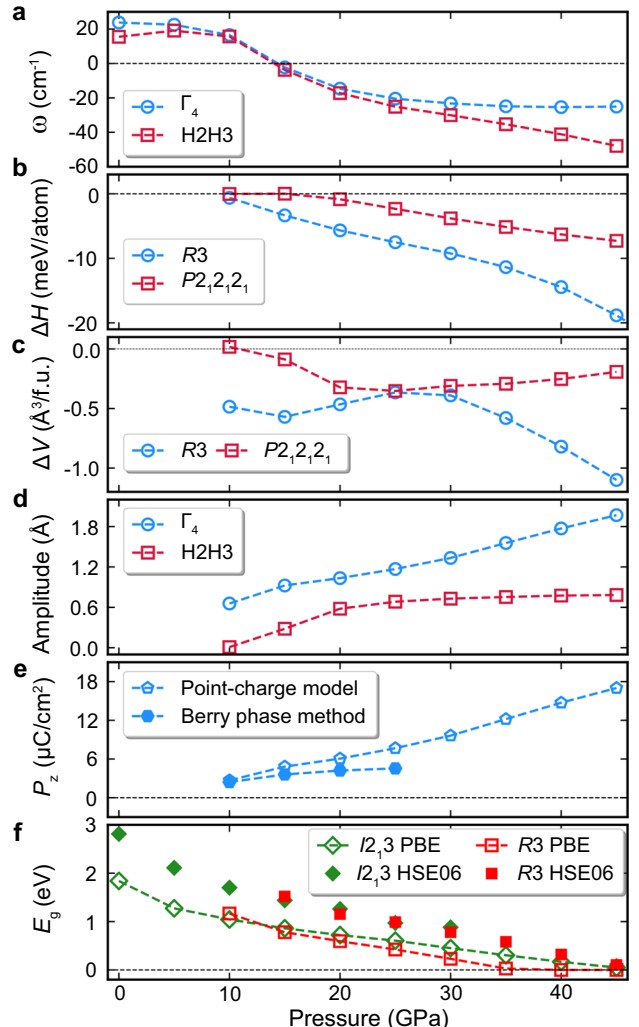

**Fig. 6 Evolution of crystal structures and physical properties of Hg₃Te₂Cl₂ as a function of pressure.** **a** Frequency of ferroelectric mode $\Gamma_4$ of $I2_13$ phase of $Hg_3Te_2Cl_2$. **b** Enthalpy difference ($\Delta H$) of $R3$ and $P2_12_12_1$ phases with respect to cubic phase $I2_13$. **c** Volume change ($\Delta V$) of $R3$ and $P2_12_12_1$ phases with respect to the cubic phase $I2_13$. **d** Distortion amplitude of $\Gamma_4$ (connected with $R3$ phase) and H2H3 (connected with $P2_12_12_1$) modes of $I2_13$ phase of $Hg_3Te_2Cl_2$. **e** Spontaneous polarization of $R3$ phase along the $c$-axis calculated by using Berry Phase method and point charge model. **f** Band gaps of $I2_13$ and $R3$ phase of $Hg_3Te_2Cl_2$.

TeHg₃, two Hg ligands and the Te center have a large displacement, whereas the other one Hg has almost no distortion. The net polarization of $R3$ phase is along the $c$-axis, which is mainly due to the uncompensated distortion of Te and Hg along the $c$-axis. All TeHg₃ tetrahedra in the unit cell have the same distortion in the H₂H₃ mode, and the main distortion of TeHg₃ involves Te and one of Hg cations.

The subgroup structures $R3$ and $P2_12_12_1$ generated by freezing the force constant (FC) eigenvectors of the $\Gamma_4$ and $H_2H_3$ modes were fully relaxed under hydrostatic pressure of 5–45 GPa. As shown in Fig. 6, although the unstable phonon mode at $H_2H_3$ is more negative than $\Gamma_4$, the $R3$ phase has more negative enthalpy gain ($\Delta E + P\Delta V$, where $E$, $P$, and $V$ are energy, pressure, and volume, respectively), smaller volume ($\Delta V$), and larger structure distortion amplitude than the $P2_12_12_1$ phase at the same pressure conditions, which is consistent with the experimental observation of high-pressure phase $R3$. The volume collapse and the slope change of $\Delta H(P)$ curve at the transition pressure indicates the

$I2_13$ to $R3$ transition is first order (Figs. 1b and 6c), which is consistent with the experiments. Our theoretical analysis shows that this is a displacive phase transition. Such a small volume collapse ($\Delta V$) is most likely due to the lattice strain effects associating with the ferroelectric-like transition, which could drive a second-order transition into a first order, as commonly observed in ferroelectric materials[53,54]. We also calculated the spontaneous polarization of the $R3$ phase and band gaps of the ambient-pressure phase $I2_13$ and high-pressure phase $R3$ as a function of pressure (see Fig. 6). Polarization of the $R3$ phase increases monotonously with increasing pressure, whereas band gaps of $I2_13$ and $R3$ phases decrease gradually. Since the PBEsol band gap at high pressure ($P \geq 30$ GPa) is too small to calculate the polarization using the Berry phase method, the point charge model (PCM) is used to estimate the polarization as well. At low-pressure region, the PCM polarization using formal charges (Hg: +2; Te: −2; Cl: −1) is nearly the same as that calculated by using the Berry phase method[55,56], indicating that the system is more ionic and the ferroelectric distortion is mainly driven by crystal structure distortion. The evolution of electronic structures of $R3$ phases of $Hg_3Te_2Cl_2$ and $Hg_3Te_2Br_2$ with pressure is shown in the Supplementary Fig. 11. With increasing pressure, the band gap between R and $\Gamma$ points closes with both valence band and conduction band crossing the Fermi level. Therefore, the carriers in these compounds at high pressure are both electrons and holes. Although PBEsol underestimates the band gaps, the shape of band structures calculated by PBEsol is very similar to HSE06, see Supplementary Fig. 12. Note that we only show the band structures along R–$\Gamma$ of $Hg_3Te_2Br_2$ due to the heavy computational cost of HSE06. Therefore, the main difference between these functionals is the critical pressure for semiconductor-to-metal transition. The band structure of the $R3$ phase at 45 GPa is similar to the recently discovered ferroelectric semimetal WTe₂ at ambient condition, where the valence band maximum and conduction band minimum cross the Fermi level just in the $\Gamma$–X direction and small hole and electron pockets are formed[24]. In $Hg_3Te_2X_2$, the size of the hole and electron pockets can be adjusted by pressure (see Supplementary Fig. 11). Also, the maximum energy barrier between two ferroelectric states with opposite polarization (19 meV/atom, see Fig. 6b) is close to that of WTe₂ (23 meV/atom). These results indicate that the orientation of polarization in metallic $Hg_3Te_2X_2$ is switchable if the pressure is properly applied on a sufficiently thin sample.

To examine the mechanism of the ferroelectric phase transition, we monitored the enthalpy gain, bond lengths, and bond interactions of the $I2_13$ to $R3$ transition <30 GPa (Supplementary Fig. 13). Enthalpy gains increase with $\Gamma_4$ distortion which indicates a spontaneous ferroelectric phase transition from $I2_13$ to $R3$. The process is associated with changes of the Hg–Te and Hg–Cl bond lengths. With the polar distortion enlarging, the bond lengths of five pairs (four Hg–Cl and one Hg–Te) are increasing, five pairs (two Hg–Cl and three Hg–Te) are decreasing, and six pairs remain almost unchanged. The Hg2–Te3 bond exhibits the largest bond increase followed by Hg2–Cl2 bond, whereas the largest decrease happens to the Hg2–Cl3 bond. The integrated COHP (-ICOHP) reflects the bond strength between atom pairs and we can see that four Hg–Te (Hg1–Te1, Hg1–Te3, Hg2–Te2, and Hg2–Te4) bonds are much stronger than other bonds and are less sensitive to the $\Gamma4$ distortion because their bond lengths almost remain unchanged. The main bond strength enhancement under the $\Gamma4$ distortion happens to the Hg1–Te2 and Hg2–Te1 bonds.

To further shed light on the origin of the pressure-induced ferroelectric-like phase transition in $Hg_3Te_2Cl_2$, we compare the pressure dependence of its FCs with that of the archetypal ferroelectric material BaTiO₃, whose ferroelectric distortion can be understood by the imaginary phonon at $\Gamma$ point[57] and is

suppressed at low pressure[9,10,58]. As shown in Supplementary Fig. 14, for $BaTiO_3$, hydrostatic pressure suppresses the ferroelectric unstable phonon mode $\Gamma_4^-$ of the cubic phase immediately, consistent with the low critical pressure observed in experiments[10,58]. The on-site and interatomic FCs of $BaTiO_3$ are calculated by displacing the symmetry equivalent atoms of the $\Gamma_4^-$ mode 0.01 Å according to the symmetry-adapted mode[59]. Positive and negative on-site FCs indicate the energy increases and decreases, respectively, when the atoms are displaced along the FC eigenvectors. As shown in Supplementary Fig. 14b–e, all the on-site FCs are positive and are increasing linearly with pressure, therefore the displaced atoms tend to be forced back to their original positions. On the other hand, positive interatomic FCs indicate the energy will be decreased if two atoms (groups) are moving in opposite directions. The polar distortion of $BaTiO_3$ is mainly due to the antiparallel displacement of Ti and $O_\parallel$ (the out-of-plane oxygen of the octahedra). At ambient pressure, the FCs on $O_\parallel$ are positive when Ti (Supplementary Fig. 14c) is displaced (Supplementary Fig. 14d). However, the interatomic FCs decrease rapidly under compression and become negative, leading to the suppression of ferroelectric instability >10 GPa. This behavior in $BaTiO_3$ is sharply in contrast to what is observed in $Hg_3Te_2Cl_2$. As shown in Supplementary Fig. 15, the FCs (Supplementary Fig. 15d, h) between Hg4 group and Te3 and Te4 groups are the dominant components of the $\Gamma_4$ mode. At ambient pressure, the positive FCs between Hg4 (six Hg atoms) group and Te3 (three of four Te atoms) group are compensated by the negative FCs between Hg4 and Te1 (four Te atoms), and the system has no unstable $\Gamma_4$ phonon mode. However, when pressure is increased, the FCs between Hg4 and Te3 increase faster than that of Hg4 and Te1 and the ferroelectric instability is established as a consequence. The same scenario is found in the case of the Te3 displacement (Supplementary Fig. 15h): the positive FCs between Te3 and Hg4 are canceled by that between Te3 and Hg2, and the increase of the FCs between Te3 and Hg4 is faster than that between Te3 and Hg2. Therefore, the unusual pressure-induced ferroelectric-like phase transition in $Hg_3Te_2X_2$ is mainly due to the special crystal structure. Phonon calculations for $I2_13$ structure of $Hg_3S_2Cl_2$ and a hypothetical compound $Hg_3Se_2F_2$ show that these compounds have similar polar phonon mode softening at 20 GPa (see Supplementary Fig. 16), indicating a general phenomenon of pressure-induced ferro-electric-like phase transition in this family of materials.

## Discussion

In this work, we have shown two defect antiperovskites $Hg_3Te_2X_2$ (X = Cl, Br) undergo uncommon ferroelectric-like structural transitions under high pressure. The underlying mechanism that leads to this unexpected noncentrosymmetric/nonpolar to non-centrosymmetric/polar transformation in these two materials is elucidated by detailed DFT calculations. We also established that the quasi-hydrostatic (nonuniform stress) condition facilitates the transition to occur at lower pressure, whereas good hydrostaticity (in He) leads to a delayed transformation. The distinct piezo-chromism, band gap closure and gradual photoconductivity increase demonstrate prominent compression-induced distinct electronic changes in these two semiconductors. The $R3$ phase of $Hg_3Te_2Br_2$ is remarkably stable over a broad pressure range and transforms to a polar metal >42.0 GPa. This study sheds light on the interplay between pressure-induced electronic interactions and ferroelectric properties of perovskite-related materials and demonstrates the possibility that polarization could be induced and enhanced by pressure in this class of materials.

## Methods

**Synthesis of $Hg_3Te_2X_2$ (X = Br, Cl) single crystals.** The chemicals Hg metal (99.999%, Alfa Aesar), $HgX_2$ (99.99%, Alfa Aesar), and Te shot (99.999%, Alfa Aesar) were purchased and used without further purification. Caution: mercury and its related compounds are toxic and great care should be taken. HgTe was prepared using Hg metal and Te shot with a stoichiometric ratio of 1:1. These two chemicals were putting into a small silica crucible and then flame-sealed in high vacuum of $\sim 10^{-4}$ mbar in a larger silica tube (13 mm for outer diameter, ~35 cm long). The sealed tube with the reaction side placed in a tube furnace and another side extended beyond the hot zone of the furnace. After heated at 520 °C for 72 h, HgTe was formed and condensed at the cold end. HgTe and $HgX_2$ (molar ratio of 2:1) mixture was placed in a silica tube and sealed under $\sim 10^{-4}$ mbar. The tube was heated to 570 °C within 12 h, kept at 570 °C for 96 h, and then slowly cooling to room temperature with the rate of 5 °C/h. Small yellow $Hg_3Te_2X_2$ single crystals were obtained by washing the reacted mixture with methanol–ethanol mixtures. Phase purity and stability were examined by powder XRD, and the titled compounds remained stable in the air for several months.

**High-pressure X-ray diffraction measurements.** For the high-pressure XRD measurements, symmetric diamond anvil cell (DAC) or Boehler-Almax plate DAC were used to generate high pressure. The ruby fluorescence method was used to determine the pressure, and the pressure was checked before and after each dif-fraction data collection[60].

Initially, single crystals of $Hg_3Te_2Cl_2$ (~51 × 35 × 25 μm³ in size) and $Hg_3Te_2Br_2$ (~30 × 20 × 10 μm³ in size) were separately loaded in two plate DACs for the high-pressure single-crystal XRD measurements. He was employed as the PTM. The single-crystal X-ray data of $Hg_3Te_2Cl_2$ were collected up to 16.1 GPa, above which only powder XRD data were recorded. For $Hg_3Te_2Br_2$, the X-ray data were collected to the experimental limit of 32.3 GPa. Then the pressure was released slowly, we found the phase transition and sample color were completely recovered when pressure released to the ambient pressure.

In another series of measurements, MEW mixture (volume ratio: 16:3:1) was used as the PTM with the purpose to explore the effect of hydrostatic condition on the phase stabilities of $Hg_3Te_2X_2$ (X = Cl, Br) and proper comparison with electrical conductivity results[44]. Single crystals with dimensions (~35 × 24 × 15 μm³ in size) were loaded in the DACs. Diffraction data were collected up to 18.2 and 32.3 GPa for $Hg_3Te_2Cl_2$ and $Hg_3Te_2Br_2$, respectively. Finally, the sample was decompressed gradually to ambient pressure. We found the initial ambient-pressure color was recovered.

The single-crystal XRD measurements were conducted at beamline 13-BM-C, GSECARS of the Advanced Photon Source (APS), Argonne National Laboratory (ANL). The X-ray wavelength ($\lambda$) equals to 0.4325 Å. Initially, the DACs was mounted at four different detector positions with all the diffraction data recorded and these data were finally merged together. A Mar165 charged-coupled device detector is used for the diffraction data collection. The exposure time was set as 1 s/ degree and the collected diffraction images were analyzed using the ATREX IDL software package[61]. A series of corrections including polarization, Lorentz, and empirical determined diamond absorption corrections were applied to the measured diffraction peaks to calculate structure factors. The unit-cell dimensions and orientation matrix were determined in RSV for each dataset and then the lattice parameters were refined, using a least squares fitting procedure. The known ambient-pressure structures (nonpolar phase I) of $Hg_3Te_2X_2$ were used as the starting model to carry out the refinements for all the high-pressure data with the aid of SHELXL-97 (ref. [62]). According to the high-pressure polar phase II, the direct methods (SHELXS-97) are used to solve the new structure. Due to the low completeness of the diffraction data of phase II (>20 GPa), the anisotropic factors $U_{ij}$ of few Cl, Br, and Te atoms were restrained to approximate isotropic shape by using command ISOR 0.01 of SHELXL-97 (ref. [62]).

In order to check the phase stabilities of $Hg_3Te_2X_2$ samples under different hydrostatic conditions, high-pressure powder X-ray measurements were conducted without pressure medium. Both fine grounded polycrystalline samples were loaded, and pressurized in plate and symmetric DACs. The diffraction data were collected at 16-ID-B beamline of the High Pressure Collaborative Access Team (HPCAT) at APS, ANL ($\lambda = 0.4066$ Å). The diffraction images were integrated using the *Dioptas* program[63]. The powder XRD data were analyzed by Le Bail fitting method using GSAS-EXPGUI package[64]. The single-crystal X-ray structures of $Hg_3Te_2X_2$ were used as the starting models to carry out all the powder data refinements.

**Electrical transport and photoconductivity measurements.** Electrical transport measurements of $Hg_3Te_2X_2$ (X = Cl, Br) were performed by the quasi-four ter-inal technique. A mixture of fine alumina and epoxy was used to insulate Pt electrodes from the DAC and the stainless steel gasket. Four Pt electrodes (5 μm thick) were arranged to contact the $Hg_3Te_2X_2$ sample and connecting to the external copper wires (0.25 mm diameter). The DAC was inserted in a closed-cycle He cryostat with temperature varying from 296 to 5 K. A Keithley 2400 Source-Meter was employed to generate direct current in the circuit and measure voltage across the sample. For lower resistance values (<5000 Ω), we used AC technique using Stanford Research SR830 digital lock-in amplifier at frequency of 17.777 Hz. Photoconductivity measurements of $Hg_3Te_2Br_2$ were conducted by the two-probe

method using visible 532 nm green light illuminating the sample under metallurgical microscope.

**DFT calculations**. All DFT calculations are performed using the projector augmented-wave method[65,66], as implemented in the Vienna Ab initio Simulation Package[67,68]. We use the PBEsol version of generalized gradient approximation functional[69], a $\Gamma$-centered $K$-points grid of $10 \times 10 \times 10$, and a plane wave basis set with 520 eV cutoff energy to relax the crystal structures and compute the electronic structures of the nonpolar and polar phases. Phonon dispersion of the $I2_13$ phase is calculated by using the finite displacement method as implemented in the Phonopy code[70], where a $2 \times 2 \times 2$ supercell and a $\Gamma$-centered $K$-points grid of $3 \times 3 \times 3$ are used. The electronic spontaneous polarization ($P_s$) of the ferroelectric phase ($R3$) is calculated using the Berry phase method[55,56]. The polarization of formal charge is calculated from the formal charge ($Z$) and atom displacements ($u$) of the polar structure ($R3$) with respect to the reference phase ($I2_13$) as $P_\alpha = \frac{e}{\Omega} \sum_{k,\beta} Z^*_{k,\alpha\beta} u_{k,\beta}$, where $\Omega$ and $e$ are the volume of the unit cell and the elementary charge, respectively.

## Data availability
The data that support the findings of this study are available from the corresponding author upon reasonable request. The X-ray crystallographic data for structures reported in this paper have been deposited in the Cambridge Crystallographic Data Center (CCDC), under reference numbers CCDC 2021956–2021966 and 2021970–2021975. These data can be obtained free of charge from the Cambridge Crystallographic Data Center via www.ccdc.cam.ac.uk/data_request/cif.

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

## Acknowledgements

The authors would like to acknowledge Dr. Haozhe Liu (HPSTAR) for allowing us to use the gearbox for helium loading and Dr. Przemysław Dera (University of Hawaii at Manoa) for his kind demonstration of the ATREX software. We thank Dr. Jin-Ke Bao for valuable discussions, Dr. Sergy Tkachev for the assistance of helium gas loading, and Dr. Jesse Smith for experimental support of X-ray measurements. The high-pressure single crystal and powder X-ray diffraction data were respectively collected at 13-BM-C of GeoSoilEnviroCARS (The University of Chicago, Sector 13) and 16-ID-B of HPCAT (Sector 16), Advanced Photon Source (APS), Argonne National Laboratory. GeoSoilEnviroCARS is supported by the National Science Foundation-Earth Sciences (EAR-1634415), and Department of Energy-GeoSciences (DE-FG02-94ER14466). HPCAT operations are supported by DOE-NNSA's Office of Experimental Sciences. Use of the COMPRES-GSECARS gas loading system and PX2 was supported by COMPRES under NSF Cooperative Agreement EAR-1661511, and by GSECARS through NSF grant EAR-1634415 and DOE grant DE-FG02-94ER14466. Work at Argonne (sample preparation, characterization, and crystal growth) is supported by the U.S. DOE, Office of Basic Energy Science, Materials Science and Engineering Division. Use of the Advanced Photon Source at Argonne National Laboratory was supported by the U.S. Department of Energy, Office of Science, Office of Basic Energy Sciences, under Contract No. DE-AC02-06CH11357. This work at University of Utah was supported by the U.S. Department of Energy, Office of Science, Fusion Energy Sciences under Award Number DE-SC0020340 (S.D.). DFT calculations at Northwestern University acknowledge National Science Foundation through the MRSEC program (NSF-DMR 1720139) at the Materials Research Center. The authors acknowledge the computing resources provided by the National Energy Research Scientific Computing Center (NERSC), a U.S. Department of Energy Office of Science User Facility operated under Contract No. DE-AC02-05CH11231.

## Author contributions

S.D. and M.G.K. conceived the project. W.C. and S.D. designed the experiments; H.L., D.Y.C. and M.G.K. synthesized single-crystal samples; W.C., R.Z., T.B. and S.D. collected the high-pressure X-ray diffraction data with the assistance of D.Z.; W.C. analyzed the X-ray data and conducted high-pressure electrical transport measurements; and J.H. and C.W. carried out the DFT calculations. W.C. and J.H. wrote the first draft of manuscript and commented by S.D., M.G.K. and C.W.; and all other authors contributed to editing the manuscript thereafter.

## Competing interests

The authors declare no competing interests.
