## [Peer Review File · Nature Communications]

REVIEWER COMMENTS

Reviewer #1 (Remarks to the Author):

The authors performed a comprehensive study (both in experiments and in DFT calculations) on defect antiperovskites $\text{Hg}_3\text{Te}_2\text{X}_2$ ($\text{X} = \text{Cl}, \text{Br}$). They experimentally found a noncentrosymmetric-nonpolar to noncentrosymmetric-polar structural phase transition in both compounds around 15-18 GPa. Under 42 GPa, $\text{Hg}_3\text{Te}_2\text{Br}_2$ undergoes a semiconductor-to-metal transition and a polar metallic state is induced.

The analysis is very detailed (including a long supplementary material), which I appreciate. The discovery of emergent polarization induced by pressure is very interesting and also rare, which definitely deserves publication in some form. However, I have a few comments/concerns for the authors to address, which (I hope) may improve the overall quality of the current work.

1. The authors experimentally found that in both compounds, there is a first-order (discontinuous) structural phase transition from $I2_13$ to $R3$ (Fig. 1a). However, from the theoretical calculation, it seems that ΔH between $R3$ and $I2_13$ changes gradually from zero to negative around 10 GPa (Fig. 6b). Does that mean $R3$ structure can not be stabilized under a pressure (smaller than 10 GPa)? If that is the case, it seems like a second-order phase transition in DFT calculations? Could the author comment on that?

2. The authors calculated the phonon spectrum of the $I2_13$ crystal structure. Under high pressure, they found that the phonon spectrum exhibits imaginary phonon modes. The imaginary phonon mode at Gamma point corresponds to a polar mode. The calculations by themselves are correct. However, when the authors link the imaginary phonon mode to the $R3$ polar structure, care is needed. The concern is that from experiment the noncentrosymmetric-nonpolar to noncentrosymmetric-polar phase transition is first-order rather than second-order. The authors made an analogy to BaTiO_3 , in which the cubic structure also exhibits an imaginary phonon mode from G to X to M (PRB 60, 836). But BaTiO_3 exhibits a second-order phase transition from cubic to tetragonal structure (at least from the viewpoint of a displacive model). If the Gamma_4 imaginary phonon in the $I2_13$ structure is related to the polar distortions in the $R3$ structure, then should one expect that the structural phase transition would be second-order rather than first-order?

3. The authors mentioned "polar metal" both in the title and in the abstract. Intrinsic polar metals are indeed rare and interesting. However, experimentally the authors convincingly show that under 42 GPa, the resistance of $\text{Hg}_3\text{Te}_2\text{Br}_2$ shows a metallic behaviour from 0 to 300K (Fig. 3b). And $\text{Hg}_3\text{Te}_2\text{Br}_2$ persists in the $R3$ polar structure. An intrinsic polar metal should exhibit a continuous centrosymmetric-to-polar phase transition upon cooling. So under 42 GPa, if one increases the temperature, is it possible to find a corresponding centrosymmetric structure in $\text{Hg}_3\text{Te}_2\text{Br}_2$? Theoretically, DFT calculations show that using PBE, the gap of the $R3c$ structure is closed above 35 GPa. First, I am wondering how the authors calculated Born effective charges at 45 GPa when the gap is closed? I would natively think Born effective charges are only well-defined in insulators. Second, before the gap is closed, why did the authors not use the Berry phase method to directly calculate the polarization (at least the polarization change as a function of pressure)?

4. Also related to the above comment, could the authors show the DFT-calculated electronic structure above 40 GPa (when the gap is closed)? What is the shape of the Fermi surface and whether the carriers are dominantly electrons or holes or mixed? This information might be helpful for further

transport measurements.

The remaining comments are minor technical points, which are easy to fix:

5. While this does not affect my evaluation, the figure resolution in the current version is very low. In some figures (such as Fig. 5), it is difficult to see clearly all the details (after zooming in).

6. About the presentation:

a) The caption of Fig. 5: Phonon dispersion of R3 phase of $\text{Hg}_3\text{Te}_2\text{Cl}_2$ under pressure. I think it should be "I2_13 phase". The R3 phase is stable under pressure, correct? Or did I miss something here?

b) Fig. 6 discusses a few different crystal structures of $\text{Hg}_3\text{Te}_2\text{Cl}_2$. It is better to specify which structure is used. For example, "Frequency of ferroelectric mode Γ_4 of $\text{Hg}_3\text{Te}_2\text{Cl}_2$ ", is this of I2_13 structure?

c) Page 11, "in contrast to what observed in $\text{Hg}_3\text{Te}_2\text{Cl}_2$." should be "in contrast to what is observed in $\text{Hg}_3\text{Te}_2\text{Cl}_2$ ".

d) In the conclusion, "demonstrates the possibility that ferroelectricity could be induced and enhanced by pressure in this class of materials." I suggest the authors that "ferroelectricity" is replaced by "polarization" or "a polar state". Strictly speaking, ferroelectricity not only means polarization but also switchability of a polarization. As far as I understand, the current study does not demonstrate switching the polarization of $\text{Hg}_3\text{Te}_2\text{X}_2$ ($\text{X}=\text{Cl}$ or Br), probably hindered by a high pressure environment. Of course, if the authors can demonstrate a polarization switching, they can definitely keep the word "ferroelectricity" in the conclusion and the results will be more desirable.

Reviewer #2 (Remarks to the Author):

Ferroelectric metal is one of the most interesting emergent phenomena proposed in 1960s, which has drawn considerable attention since the discoveries in LiOsO_3 . In this paper, the authors extensively studied pressure-dependent evolution of the crystal structure, resistance, electronic structures of two defected antiperovskites $\text{Hg}_3\text{Te}_2\text{X}_2$ ($\text{X} = \text{Cl}, \text{Br}$), and claimed ferroelectric-like transitions to polar metal states under high pressure. The phase transitions and lattice parameter variation were unambiguously unveiled by in situ diffraction techniques. Theoretical calculations on the dynamics feature elucidated that the phonon mode evolution under pressure plays a critical role on the observed structural transitions. Overall, this is a solid work and worthy of publishing after a thorough revision.

1. The resistance does increase with increasing temperature at 42.3 GPa, showing metallic behavior. Accordingly, the authors claimed a band gap closing from ~ 2.5 to 0 eV in the abstract. However, from Fig. 3b, one cannot rule out any possible bad metal state. Presumably, the slope of resistance-T plot will keep changing at higher pressure. Moreover, first-principles DFT calculations of the DOS usually give smaller band gap than the real case. So it is too brave to claim a metal state unless further validation can manifest this. Otherwise, a band gap closing from ~ 2.5 eV to a metallic like state could be fine.

2. The current form of Fig. 3b is not clear enough to show the pressure and temperature dependent

transport behavior. Fitting of the data in $\ln\rho - 1/T^n$ should be applied to analyze the conduction mechanism (Mott VRH, Fermi liquid, etc). A temperature-pressure phase diagram is a plus, with conductivity illustrated by color code or phase boundary.

3. The author should have done a more comprehensive literature screening. For example, significant experimental validation on polar metal has been recently reported in the van der Waals compound WTe₂ (P. Sharma et al, A room - temperature ferroelectric semimetal, Sci. Adv. 2019, 5, eaax5080), which is a room temperature ferroelectric Weyl semimetal. In contrast to LiOsO₃, the ferroelectric state in WTe₂ is experimentally proved. The authors should compare the discoveries in this manuscript with what reported for WTe₂.

4. What is lowest temperature in resistance measurements? The answer of this question relates to possible quantum critical point around the transition area.

5. The paper can be further improved by careful proof-reading: "three-dimensional" should be defined at its first presence only and shorted for "3D" to meet the standard phrase; "9.3318 Å and 9.5866 Å" should be "9.3318 and 9.5866 Å".

6. The current figure resolution is totally unacceptable.

Responses to the Reviewers

We thank the reviewers for their insightful comments and suggestions. Below, we present our point-by-point responses to each reviewer's comments.

Reviewer #1

The authors performed a comprehensive study (both in experiments and in DFT calculations) on defect antiperovskites $\text{Hg}_3\text{Te}_2\text{X}_2$ ($\text{X} = \text{Cl}, \text{Br}$). They experimentally found a noncentrosymmetric-nonpolar to noncentrosymmetric-polar structural phase transition in both compounds around 15-18 GPa. Under 42 GPa, $\text{Hg}_3\text{Te}_2\text{Br}_2$ undergoes a semiconductor-to-metal transition and a polar metallic state is induced. The analysis is very detailed (including a long supplementary material), which I appreciate. The discovery of emergent polarization induced by pressure is very interesting and also rare, which definitely deserves publication in some form. However, I have a few comments/concerns for the authors to address, which (I hope) may improve the overall quality of the current work.

Response: We are grateful to the reviewer for the positive assessment of our work and detailed review of our manuscript. In the revised manuscript we added more specific discussions based on the reviewer's suggestions.

1. The authors experimentally found that in both compounds, there is a first-order (discontinuous) structural phase transition from $I2_{13}$ to $R3$ (Fig. 1a). However, from the theoretical calculation, it seems that ΔH between $R3$ and $I2_{13}$ changes gradually from zero to negative around 10 GPa (Fig. 6b). Does that mean $R3$ structure can not be stabilized under a pressure (smaller than 10 GPa)? If that is the case, it seems like a second-order phase transition in DFT calculations? Could the author comment on that?

Response: We thank the reviewer for bringing up this point. According to the experimental data the phase transition from $I2_{13}$ to $R3$ is first-order. However, the volume collapse (ΔV) during the phase transition is subtle ($\sim 0.3\%$ for $\text{Hg}_3\text{Te}_2\text{Br}_2$ and $\sim 0.5\%$ for $\text{Hg}_3\text{Te}_2\text{Cl}_2$, see Figure 1b of the main text). Such a small volume drop is most likely due to the strain effects that originate from differences between the lattices of nonpolar and polar phases. It is known that lattice strain effect associated with ferroelectric phase transition could drive a second-order phase transition into a first-order type [please see references: Rabe, K.M., Ahn, C.H. & Triscone, J.M. *Physics of Ferroelectrics: A Modern Perspective*. Springer Berlin Heidelberg, 2007; Kumar, A. & Waghmare, U.V. First-principles free energies and Ginzburg-Landau theory of domains and ferroelectric phase transitions in BaTiO_3 . *Phys. Rev. B* 82, 054117 (2010).].

In the first-order phase transitions induced by pressure, both changes of volume (ΔV) and energy (ΔE) are discontinuous, however, the change in enthalpy (ΔH) is continuous. This is different from temperature-induced first-order phase transitions where both changes of entropy (ΔS) and enthalpy (ΔH) are discontinuous. Instead, in cases of pressure-induced first-order transition, there is a change in the slope of the $\Delta H(P)$ curve at the transformation point. In our theoretical simulations, we do observe a small slope change and volume drop between these two phases (Figure 6b and 6c of the manuscript), and therefore this phase transition is first-order, which is consistent with experiments.

The reviewer is correct, the $R3$ structure cannot be stabilized if the pressure is less than 10 GPa, *i.e.*, the $R3$ relaxed to $I2_13$ essentially. The emergence of $R3$ phase is due to the unstable phonon mode Γ_4 . As we explained above, this does not necessarily mean it is a second-order phase transition. The strain effect has to be considered to fully understand the phase transition type theoretically [please see reference: Kumar, A. & Waghmare, U.V. First-principles free energies and Ginzburg-Landau theory of domains and ferroelectric phase transitions in BaTiO_3 . *Phys. Rev. B* 82, 054117 (2010).].

We added the following sentences in the revised manuscript:

“The volume collapse and the slope change of $\Delta H(P)$ curve at the transition pressure indicate the $I2_13$ to $R3$ transition is first-order (Figs. 1b and 6c), which is consistent with the experiments. Our theoretical analysis shows that this is a displacive phase transition. Such a small volume collapse (ΔV) is most likely due to the lattice strain effects associating with the ferroelectric-like transition, which could drive a second-order transition into a first-order, as commonly observed in ferroelectric materials^{57,58}.”

Figure 6 is revised: Our original plot of Figure 6 has shown that the $R3$ structure is relaxed back to $I2_13$ at low-pressure range (these two structures have the same values, such as ΔH). However, we realized this makes it more confused. For clarity, therefore, we removed the points where the $R3$ structure could not be stabilized (< 10 GPa) in the revised manuscript.

New references added:

57. Rabe, K.M., Ahn, C.H. & Triscone, J.M. *Physics of Ferroelectrics: A Modern Perspective*. (Springer Berlin Heidelberg, 2007).
58. Kumar, A. & Waghmare, U.V. First-principles free energies and Ginzburg-Landau theory of domains and ferroelectric phase transitions in BaTiO_3 . *Phys. Rev. B* 82, 054117 (2010).

Fig. 6 Evolution of crystal structures and physical properties of $\text{Hg}_3\text{Te}_2\text{Cl}_2$ as a function of pressure. **a** Frequency of ferroelectric mode Γ_4 of $I2_13$ phase of $\text{Hg}_3\text{Te}_2\text{Cl}_2$. **b** Enthalpy difference (ΔH) of $R3$ and $P2_12_12_1$ phases with respect to cubic phase $I2_13$. **c** Volume change (ΔV) of $R3$ and $P2_12_12_1$ phases with respect to the cubic phase $I2_13$. **d** Distortion amplitude of Γ_4 (connected with $R3$ phase) and $H2H3$ (connected with $P2_12_12_1$) modes of $I2_13$ phase of $\text{Hg}_3\text{Te}_2\text{Cl}_2$. **e** Spontaneous polarization of $R3$ phase along the c -axis calculated by using Berry Phase method and point charge model. **f** Bandgaps of $I2_13$ and $R3$ phase of $\text{Hg}_3\text{Te}_2\text{Cl}_2$.

2. The authors calculated the phonon spectrum of the $I2_13$ crystal structure. Under high pressure, they found that the phonon spectrum exhibits imaginary phonon modes. The imaginary phonon mode at Gamma point corresponds to a polar mode. The calculations by themselves are correct. However, when the authors link the imaginary phonon mode to the $R3$ polar structure, care is needed. The concern is that from experiment the noncentrosymmetric-nonpolar to noncentrosymmetric-polar phase transition is first-order

rather than second-order. The authors made an analogy to BaTiO₃, in which the cubic structure also exhibits an imaginary phonon mode from G to X to M (PRB 60, 836). But BaTiO₃ exhibits a second-order phase transition from cubic to tetragonal structure (at least from the viewpoint of a displacive model). If the Gamma_4 imaginary phonon in the I2_13 structure is related to the polar distortions in the R3 structure, then should one expect that the structural phase transition would be second-order rather than first-order?

Response: We thank the reviewer for pointing this out and we fully understand the reviewer's concerns. In the ferroelectric phase transition, it is common to observe coupling between the polarization and the lattice strain. The strain associated with the lattice constant (volume) difference between two phases could drive a second-order phase transition into a first-order phase transition, as observed in BaTiO₃ [Rabe, K.M., Ahn, C.H. & Triscone, J.M. *Physics of Ferroelectrics: A Modern Perspective*. Springer Berlin Heidelberg, 2007; Kumar, A. & Waghmare, U.V. First-principles free energies and Ginzburg-Landau theory of domains and ferroelectric phase transitions in BaTiO₃. *Phys. Rev. B* 82, 054117 (2010)]. Experimentally, the ferroelectric phase transition of BaTiO₃ (also PbTiO₃ and many other perovskites) is found to be the first-order (with small volume discontinuity) [M. Acosta et. al., *Applied Physics Reviews* 4, 041305 (2017); J. Rossetti et. al., *Journal of Physics: Condensed Matter* 17, 3953 (2015)]. However, these first-order phase transitions are displacive transitions, which can be first-order or second-order (*Inorganic Structure Chemistry*, Ulrich Muller, 2nd Edition, John Wiley & Sons, Ltd). In general, the soft-mode theory can be used to understand the main features of the displacive phase transition that satisfies the group and subgroup relation. As mentioned by the reviewer, the displacive phase transition of BaTiO₃ has been well-understood based on the imaginary phonon mode at the Γ point, which is very similar to the case we found in Hg₃Te₂X₂.

We emphasize that our DFT calculations at 0 K cannot capture the whole picture of the first-order displacive phase transition in Hg₃Te₂X₂, which is similar to the archetypal ferroelectric materials BaTiO₃ and PbTiO₃. As demonstrated in previous studies [Kumar and Waghmare, *Phys. Rev. B* 82, 054117 (2010); Pitike et.al., *J. Mater. Sci.* 54, 8381 (2019)], the simulation at finite temperature using effective Hamiltonian which includes the strain effects can reproduce the first-order phase transition observed in experiments. However, this method is too complicated for our system and is beyond the scope of this paper. Here, we are aimed to understand the origin of the ferroelectric distortion.

New reference added:

61. Ghosez, P., Cockayne, E., Waghmare, U.V. & Rabe, K.M. Lattice dynamics of BaTiO₃, PbTiO₃, and PbZrO₃: A comparative first-principles study. *Phys. Rev. B* **60**, 836-843 (1999).

3. The authors mentioned "polar metal" both in the title and in the abstract. Intrinsic polar metals are indeed rare and interesting. However, experimentally the authors convincingly show that under 42 GPa, the resistance of Hg₃Te₂Br₂ shows a metallic behaviour from 0 to 300K (Fig. 3b). And Hg₃Te₂Br₂ persists in the R3 polar structure. An intrinsic polar metal should exhibit a continuous centrosymmetric-to-polar phase transition upon cooling. So under 42 GPa, if one increases the temperature, is it possible to find a corresponding centrosymmetric structure in Hg₃Te₂Br₂?

Response: We thank the reviewer for the valuable suggestions and agree with the reviewer that a polar-to-centrosymmetric phase transition should occur in a *ferroelectric-like* structure transition with increasing temperature, as observed in LiOsO₃ [Shi et. al, *Nature Materials* 12, 1024 (2013)]. If such transition emerges in Hg₃Te₂X₂, it will be another significant achievement. However, such experiments require synchrotron X-ray studies under variable temperature and high pressure conditions, which is currently unavailable to us. However, our transport measurements below 42 GPa (Fig.3 in the main text) do not show anomalous behavior in any of the *R-T* curves between 5.0 to 300 K (e.g. phase transitions upon cooling generally related to the emergence of kinks in the *R-T* curve, see also Shi et. al, *Nature Materials* 12, 1024 (2013)). Therefore, instead of claiming observation of a *ferroelectric-like metal*, we just use the term *polar metal*. The evidence of polar metal does not necessarily require a polar-to-centrosymmetric phase transition with increasing temperature, for example, see Kim et. al., *Nature* 533, 68 (2016) and therefore, we hope using this terminology appropriately fits our observations here.

Theoretically, DFT calculations show that using PBE, the gap of the R3c structure is closed above 35 GPa. First, I am wondering how the authors calculated Born effective charges at 45 GPa when the gap is closed? I would natively think Born effective charges are only well-defined in insulators. Second, before the gap is closed, why did the authors not use the Berry phase method to directly calculate the polarization (at least the polarization change as a function of pressure)?

Response: The reviewer is correct; the Born effect charge is only well-defined in insulators. We apologize that we made a mistake in computing polarization at high pressure using Born effective charge and we are grateful to reviewer to pointing this out. In the revised manuscript (see new Figure 6), we recalculated polarization by using Berry phase method as suggested by the reviewer and the point charge model (PCM). As shown in Figure 6e, the polarization calculated by Berry phase and PCM are very close at low-pressure region. At high-pressure region, we only show the polarization calculated using the PCM since the Berry phase calculation is not reliable when the PBEsol bandgap is too small. The Berry phase polarization is slightly smaller than our previous calculations using Born effect charge because we used the Born effective charge of the reference

phase. Overall, they both have very similar trend, and our main conclusions remain the same.

We added the following sentences in the revised manuscript:

“Since the PBEsol band gap at high pressure (≥ 30 GPa) is too small to calculate the polarization using the Berry phase method, the point charge model (PCM) is used to estimate the polarization. In the low-pressure region, the PCM polarization using formal charges (Hg: +2; Te: -2; Cl: -1) is nearly the same as that calculated with the Berry phase method^{59,60}, indicating that the system is more ionic and the ferroelectric distortion is mainly driven by crystal structure distortion.”

New references added:

59. King-Smith, R.D. & Vanderbilt, D. Theory of polarization of crystalline solids. *Phys. Rev. B* **47**, 1651-1654 (1993).
60. Resta, R. Theory of the electric polarization in crystals. *Ferroelectrics* **136**, 51-55 (1992).

4. Also related to the above comment, could the authors show the DFT-calculated electronic structure above 40 GPa (when the gap is closed)? What is the shape of the Fermi surface and whether the carriers are dominantly electrons or holes or mixed? This information might be helpful for further transport measurements.

Response: We thank the reviewer for the suggestion. We have included the PBEsol calculated band structures and Fermi surface of *R3* phase of $\text{Hg}_3\text{Te}_2\text{Br}_2$ at the pressure between 30 and 45 GPa in the Supplementary Fig. 11. Although PBEsol functional underestimates bandgaps of $\text{Hg}_3\text{Te}_2\text{X}_2$, we observe a similar band shape of PBEsol and HSE06 (HSE06 slightly overestimates the bandgap, please see the Supplementary Fig. 12). With increasing pressure, the band gap between R and Γ points closes with both valence band and conduction band cross the Fermi level. Therefore, the carriers are both of electrons and holes.

We added the following sentences in the revised manuscript.

“The evolution of electronic structures of the *R3* phases of $\text{Hg}_3\text{Te}_2\text{Cl}_2$ and $\text{Hg}_3\text{Te}_2\text{Br}_2$ with pressure is shown in the Supplementary Fig. 11. With increasing pressure, the band gap between R and Γ points closes with both valence band and conduction band crossing the Fermi level. Therefore, the carriers in these compounds at high-pressure are both electrons and holes. Although PBEsol underestimates the bandgaps, the shape of band

structures calculated by PBEsol is very similar to HSE06, see Supplementary Fig. 12. Note that we only show the band structures along R- Γ of $\text{Hg}_3\text{Te}_2\text{Br}_2$ due to the heavy computational cost of HSE06. Therefore, the main difference between these functionals is the critical pressure for semiconductor to metal transition.”

Supplementary Fig. 11 PBEsol electronic band structures of $R3$ phase of $\text{Hg}_3\text{Te}_2\text{X}_2$ under different pressures. **a, b, c** and **d** are band structures of $\text{Hg}_3\text{Te}_2\text{Cl}_2$ at 30, 35, 40, and 45 GPa, respectively. **e, f, g** and **h** are band structures of $\text{Hg}_3\text{Te}_2\text{Br}_2$ at 30, 35, 40, and 45 GPa, respectively. **i, j, k** and **l** are the Fermi surfaces of $\text{Hg}_3\text{Te}_2\text{Br}_2$ at 30, 35, 40, and 45 GPa, respectively. Red and blue colors represent valence and conduction bands, respectively.

Supplementary Fig. 12 Electronic band structures of *R3* phase of $\text{Hg}_3\text{Te}_2\text{Br}_2$ calculated by using HSE06. Only the band structure along the high symmetry points R and Γ is computed due to the heavy cost of HSE06 calculation. With the pressure increasing from 35 GPa to 45 GPa, $\text{Hg}_3\text{Te}_2\text{Br}_2$ becomes semimetal and metal gradually.

The remaining comments are minor technical points, which are easy to fix: 5. While this does not affect my evaluation, the figure resolution in the current version is very low. In some figures (such as Fig. 5), it is difficult to see clearly all the details (after zooming in).

Response: We apologize for the low resolution of the figures. In our revised manuscript, we have inserted high resolution figures in both the main text and Supplementary Information.

6. About the presentation:

a) The caption of Fig. 5: Phonon dispersion of *R3* phase of $\text{Hg}_3\text{Te}_2\text{Cl}_2$ under pressure. I think it should be "I2_13 phase". The *R3* phase is stable under pressure, correct? Or did I miss something here?

Response: We thank the reviewer for pointing out this typo. The $I2_13$ phase is the correct phase. We have corrected it in the revised manuscript.

b) Fig. 6 discusses a few different crystal structures of $\text{Hg}_3\text{Te}_2\text{Cl}_2$. It is better to specify which structure is used. For example, "Frequency of ferroelectric mode Γ_4 of $\text{Hg}_3\text{Te}_2\text{Cl}_2$ ", is this of $I2_13$ structure?

Response: We thank the reviewer for the comment. We have made it clear in the revised manuscript.

c) Page 11, "in contrast to what observed in $\text{Hg}_3\text{Te}_2\text{Cl}_2$." should be "in contrast to what is observed in $\text{Hg}_3\text{Te}_2\text{Cl}_2$ ".

Response: We thank the reviewer for pointing out this typo. We have corrected it in the revised manuscript.

d) In the conclusion, "demonstrates the possibility that ferroelectricity could be induced and enhanced by pressure in this class of materials.". I suggest the authors that "ferroelectricity" is replaced by "polarization" or "a polar state". Strictly speaking, ferroelectricity not only means polarization but also switchability of a polarization. As far as I understand, the current study does not demonstrate switching the polarization of $\text{Hg}_3\text{Te}_2\text{X}_2$ ($\text{X}=\text{Cl}$ or Br), probably hindered by a high pressure environment. Of course, if the authors can demonstrate a polarization switching, they can definitely keep the word "ferroelectricity" in the conclusion and the results will be more desirable.

Response: We thank the reviewer's suggestions. The reviewer is correct, we have used "polarization" to replace the "ferroelectricity" in the Conclusion part.

Reviewer #2 (Remarks to the Author):

Ferroelectric metal is one of the most interesting emergent phenomena proposed in 1960s, which has drawn considerable attention since the discoveries in LiOsO_3 . In this paper, the authors extensively studied pressure-dependent evolution of the crystal structure, resistance, electronic structures of two defected antiperovskites $\text{Hg}_3\text{Te}_2\text{X}_2$ ($\text{X} = \text{Cl}, \text{Br}$), and claimed ferroelectric-like transitions to polar metal states under high pressure. The phase transitions and lattice parameter variation were unambiguously unveiled by in situ diffraction techniques. Theoretical calculations on the dynamics feature elucidated that the phonon mode evolution under pressure plays a critical role on the observed structural transitions. Overall, this is a solid work and worthy of publishing after a thorough revision.

Response: We appreciate the reviewer's comments and detailed review of our manuscript. In the revised manuscript we added more specific discussions based on the reviewer's suggestions.

1. The resistance does increase with increasing temperature at 42.3 GPa, showing metallic behaviour. Accordingly, the authors claimed a bad gap closing from ~ 2.5 to 0 eV in the abstract. However, from Fig. 3b, one cannot rule out any possible bad metal state. Presumably, the slope of resistance-T plot will keep changing at higher pressure. Moreover, first-principles DFT calculations of the DOS usually give smaller band gap than the real case. So it is too brave to claim a metal state unless further validation can manifest this. Otherwise, a band gap closing from ~ 2.5 eV to a metallic like state could be fine.

Response: We thank the reviewer for bringing up this point. The reviewer is correct, we have changed the 0 eV to metallic-like state in the Abstract as suggested by the reviewer and refer to the behavior as metallic-like.

Generally, PBEsol gives smaller bandgaps than experiment for $\text{Hg}_3\text{Te}_2\text{X}_2$, and therefore we also provided the bandgaps calculated by HSE06, which predicts slightly larger bandgaps. Since HSE06 calculation is very expensive, we only show the band structures along R- Γ , where the gap closes. In the revised Supplementary Figure 12, we show the HSE06 band structures of $\text{Hg}_3\text{Te}_2\text{Br}_2$ under pressure of 35, 40, 45 GPa. It is clear that closure of bandgap is observed at 40 GPa and $\text{Hg}_3\text{Te}_2\text{Br}_2$ completely transforms a metal at 45 GPa.

Supplementary Fig. 12 Evolution of electronic band structures of $\text{Hg}_3\text{Te}_2\text{Br}_2$ under pressure calculated by using HSE06. Only the band structure along the high symmetry points R and Γ is computed due to the heavy cost of HSE06 calculation. With the pressure increasing from 35 GPa to 45 GPa, $\text{Hg}_3\text{Te}_2\text{Br}_2$ becomes semimetal and metal gradually.

2. The current form of Fig. 3b is not clear enough to show the pressure and temperature dependent transport behavior. Fitting of the data in $\ln \rho - 1/T^n$ should be applied to analyze the conduction mechanism (Mott VRH, Fermi liquid, etc). A temperature-pressure phase diagram is a plus, with conductivity illustrated by color code or phase boundary.

Response: We appreciate the reviewer's valuable suggestions. Since the samples used in experiments in diamond anvil cell typically have irregular shapes which changes non-uniformly under pressure, it is extremely difficult to calculate the sample's resistivity accurately under compression. Here we report the electrical resistance R instead of the resistivity ρ .

We found our resistance data below 40.2 GPa can be well fitted to the three-dimensional (3-D) variable range hopping (VRH) model $R(T) = R_0 \exp(T_0/T)^{1/4}$ (where T_0 is the characteristic temperature) at low temperature range. These $\ln R-1/T^{1/4}$ fittings show linearity indication below 20 K, revealing the carrier conduction is dominated by the 3D VRH mechanism (see Figure R1 and in the Supplementary Fig. 8b).

It is noteworthy that we have attempted to use also one-dimensional (1D) VRH model $R(T) = R_0 \exp(T_0/T)^{1/2}$ and two-dimensional (2D) $(1/T^{1/3})$ VRH model $R(T) = R_0 \exp(T_0/T)^{1/3}$, but the 3D model provides a better fit than 1D and 2D VRH models (*i.e.* smaller residual sum of squares), so we chose 3D model here.

Figure R1. 3-D $\ln R-1/T^{1/4}$ (VRH model) at 27.2, 29.3, 34.3, 36.5, 38.3 and 40.2 GPa. The inset shows $T^{1.8}$ dependence of the resistance at pressure of 42.3 GPa and below 70 K, which is well consistent with the formula $R(T) = R_0 + AT^{1.8}$ with $R_0 = 1.76 \text{ }\Omega$ and $A = 4.32 \times 10^{-5}$.

Moreover, we also analyzed the resistance data at 42.3 GPa (below 70 K) with the powder-law formula, *i.e.*, $\rho(T) = \rho_0 + AT^n$ (for Fermi liquid $n = 2$), where ρ_0 is the residual resistivity. In our case, the resistivity ρ is replaced by resistance R and the equation changes to be $R(T) = R_0 + AT^n$. We found the metallic state of $\text{Hg}_3\text{Te}_2\text{Br}_2$ shows slight deviation from the Fermi liquid (FL) behavior with $n = 1.8$. These information are now added to the revised manuscript (see inset of Figure 1R). Pressure-induced non-Fermi liquid (NFL) to FL transition has been observed in strong correlated systems. However, $\text{Hg}_3\text{Te}_2\text{Br}_2$ is not a strongly correlated compound therefore it is unlikely that the slight deviation is due to NFL behavior.

Based on the reviewer's suggestion, we also added the P - T phase diagram (Fig. 3c) in the revised Figure 3.

Figure 3. Pressure-induced metallization of $\text{Hg}_3\text{Te}_2\text{Br}_2$. **a** Pressure-dependent electrical resistances of $\text{Hg}_3\text{Te}_2\text{Br}_2$ at room temperature. **b** Temperature dependence of the electrical resistance of $\text{Hg}_3\text{Te}_2\text{Br}_2$ at different pressures. The inset enhances the semiconductor-to-metal transition at 42.3 GPa. **c** P - T phase diagram of $\text{Hg}_3\text{Te}_2\text{Br}_2$ on the basis of resistance measurements. Solid line: semiconductor-metal phase boundary.

We added the following discussions in the main text

“...the $R(T)$ curves show negative dR/dT when pressure is below 36.5 GPa in the range of 5-300 K, indicative of semiconducting character. Between 38.3 to 40.2 GPa, the dR/dT displays positive in the high temperature region, whereas it changes to negative in the low temperature region (e.g. 53 K for 40.2 GPa). With increasing the pressure above 42.0 GPa, a positive dR/dT is observed throughout all temperature range (5-300 K), implying the complete metallization of $\text{Hg}_3\text{Te}_2\text{Br}_2$.”

“We used variable range hopping (VRH) model $R(T) = R_0 \exp(T_0/T)^{1/n}$ (where R_0 is a characteristic temperature and n is an integer (1-4) depending on the conduction

mechanism) to analyze the resistance data below 40.2 GPa⁵³. We found the 3D VRH model ($n = 4$) provides the best fitting to the resistance data below 20 K for all pressures, revealing the carrier conduction is dominated by the 3D VRH mechanism (Supplementary Fig. 8b). In order to analyze the metallic state, we tried to apply the power-formula $R(T) = R_0 + AT^n$ to fit the resistance data at 42.3 GPa below 70 K. We observed Hg₃Te₂Br₂ shows a slight deviation from ideal Fermi liquid behavior ($n = 2$) and for this pressure the $R(T)$ data can be best fitted for $n = 1.8$. Pressure-induced non-Fermi liquid (NFL) to FL transition has been observed in strong correlated systems⁵⁴. However, Hg₃Te₂Br₂ is not a strongly correlated compound therefore it is unlikely that the slight deviation is due to NFL behavior. ”

New references added:

53. Mott, N.F. *Metal Insulator Transitions*. (Taylor & Francis, London; 2004).
54. Gabáni, S. et al. Pressure-induced Fermi-liquid behavior in the Kondo insulator SmB₆: Possible transition through a quantum critical point. *Phys. Rev. B* **67**, 172406 (2003).

3. The author should have done a more comprehensive literature screening. For example, significant experimental validation on polar metal has been recently reported in the van der Waals compound WTe₂ (P. Sharma et al, A room - temperature ferroelectric semimetal, *Sci. Adv.* 2019, 5, eaax5080), which is a room temperature ferroelectric Weyl semimetal. In contrast to LiOsO₃, the ferroelectric state in WTe₂ is experimentally proved. The authors should compare the discoveries in this manuscript with what reported for WTe₂.

Response: We thank the reviewer's suggestions. We have added the mentioned reference and new discussions in the main text.

We added the following discussions in the main text:

“Recently, ferroelectricity was reported in bulk crystalline two-dimensional (2-D) WTe₂ at ambient conditions. The emergence of the ferroelectricity in this Weyl semimetal could be correlated to its layered structure together with strong electronic anisotropy²⁸.”

“The band structure of the R3 phase at 45 GPa is similar to the recently discovered ferroelectric semimetal WTe₂ at ambient pressure, where the valence band maximum and conduction band minimum cross the Fermi level just in the Γ -X direction and small hole and electron pockets are formed²⁸. In Hg₃Te₂X₂, the size of the hole and electron pockets can be adjusted by pressure (see Supplementary Fig. 11). Also, the maximum energy barrier between two ferroelectric states with opposite polarization (19 meV/atom, see Fig.

6b) is close to that of WTe_2 (23 meV/atom). These results indicate that the orientation of polarization in metallic $\text{Hg}_3\text{Te}_2\text{X}_2$ is switchable if the pressure is properly applied on a sufficiently thin sample.”

New reference added:

28. Sharma, P. et al. A room-temperature ferroelectric semimetal. *Sci. Adv.* **5**, eaax5080 (2019).

4. What is lowest temperature in resistance measurements? The answer of this question relates to possible quantum critical point around the transition area.

Response: We thank the reviewer for bringing up this interesting point. In this study, the lowest temperature in our resistance measurements was ~ 5.0 K. Also, we do not have sufficiently small increments of pressure. Therefore, it is not possible to make any conclusion about a possible quantum critical point (QCP) based on the available data. Further experiments at lower temperatures and finer pressure steps both below and above the transition pressure are necessary to investigate possibility of QCP around the transition area which can be an interesting subject for future studies.

5. The paper can be further improved by careful proof-reading: “three-dimensional” should be defined at its first presence only and shorted for “3D” to meet the standard phrase; “9.3318 Å and 9.5866 Å” should be “9.3318 and 9.5866 Å”.

Response: We appreciate the reviewer for constructive comments. We have addressed these points in the revised main text.

6. The current figure resolution is totally unacceptable.

Response: We apologize for the low resolution of the figures and thank the reviewer for bringing up this point. We have inserted high resolution figures in both the main text and supplementary information.

REVIEWERS' COMMENTS

Reviewer #1 (Remarks to the Author):

The authors performed additional calculations and satisfactorily address almost all my comments, which I appreciate. I only have one minor comment: I still believe that to claim that a material is an intrinsic polar metal, the existence of a second-order polar-to-centrosymmetric phase transition at finite temperature in a metallic environment is essential. I understand that such an experiment is not feasible to the authors and reviewers should be reasonable. However, I suggest that the authors add some text to clearly state that the second-order polar-to-centrosymmetric structural phase transition of $\text{Hg}_3\text{Te}_2\text{X}_2$ under pressure has not been confirmed directly in experiments, which warrants further investigations. This does not affect the key conclusions of the current manuscript but makes the presentation more objective and comprehensive. Other than that, I recommend the publication of this work in Nature Communications.

Reviewer #2 (Remarks to the Author):

This paper has been properly revised and can be accepted for publication after minor corrections of some typos. For instance, not all variables are set as italic.

Responses to the Reviewers

We would like to thank the reviewers for their valuable suggestions. Here, we present our point-by-point responses to each reviewer's comments.

Reviewer #1 (Remarks to the Author):

The authors performed additional calculations and satisfactorily address almost all my comments, which I appreciate. I only have one minor comment: I still believe that to claim that a material is an intrinsic polar metal, the existence of a second-order polar-to-centrosymmetric phase transition at finite temperature in a metallic environment is essential. I understand that such an experiment is not feasible to the authors and reviewers should be reasonable. However, I suggest that the authors add some text to clearly state that the second-order polar-to-centrosymmetric structural phase transition of $\text{Hg}_3\text{Te}_2\text{X}_2$ under pressure has not been confirmed directly in experiments, which warrants further investigations. This does not affect the key conclusions of the current manuscript but makes the presentation more objective and comprehensive. Other than that, I recommend the publication of this work in Nature Communications.

Response: We thank the reviewer for the valuable suggestions. Indeed it is important to distinctly clarify the findings of the manuscript. Based on the reviewer's concern, we added the following sentences in the main text.

“In LiOsO_3 at ambient pressure a broad peak appears in heat capacity, accompanied by the polar-to-centrosymmetric structural phase transition at 140 K, which presents a direct evidence of a second-order phase transition²². Such type of phase transition is also typically evidenced by a distinct anomaly in the electrical resistivity. However, in case of $\text{Hg}_3\text{Te}_2\text{X}_2$, since the polar phase emerges at high pressure, we were unable to perform similar heat capacity measurements as in LiOsO_3 . Although we could not find any evidence of anomaly in the R - T curves (Fig. 3b), this does not exclude the possibility of a second-order polar-to-centrosymmetric structural phase transition in these two compounds. Further synchrotron X-ray experiments and heat capacity measurements at variable temperature and pressure are needed to experimentally explore this possibility.”

Reviewer #2 (Remarks to the Author):

This paper has been properly revised and can be accepted for publication after minor corrections of some typos. For instance, not all variables are set as italic.

Response: We thank the reviewer for pointing these typos out. We have corrected those typos, which are marked by track changes in the main text.